



# African Easterly Waves in an Idealized General Circulation Model: Instability and Wavepacket Diagnostics

Joshua White[1] and Anantha Aiyyer[1]

[1]Department of Marine, Earth and Atmospheric Sciences, North Carolina State University

**Correspondence:** A. Aiyyer (aaiyyer@ncsu.edu)

**Abstract.**

We examine the group dynamic of African easterly waves (AEW) generated in a realistic, spatially non-homogeneous African easterly jet (AEJ) using an idealized general circulation model. Our objective is to investigate whether the limited zonal extent of the AEJ is an impediment to AEW development. We construct a series of basic states using global reanalysis fields

and initialize waves via transient heating over West Africa. The dominant response is a localized wavepacket that disperses upstream and downstream. The inclusion of a crude representation of boundary layer damping stabilizes the waves in most cases. In some basic states, however, exponential growth occurs even in the presence of damping. This shows that AEWs can occasionally emerge spontaneously. The key result is that the wavepacket in almost all cases remains within the AEJ instead of being swept away. Drawing from other studies, this also suggests that even the damped waves can grow if coupled

with additional sources of energy such as moist convection and dust radiative feedback. The wavepacket in the localized AEJ appears to satisfy a condition for absolute instability, a form of spatial hydrodynamic instability. However, this needs to be verified more rigorously. Our results also suggest that the intermittent nature of AEWs is mediated, not by transitions between convective and absolute instability, but likely by external sources such as propagating equatorial wave modes.

## 1  Introduction

African easterly waves (AEWs) are the dominant synoptic-scale feature of the summer-time West African monsoon (WAM). The AEW stormtrack extends across the southern and northern sides of the mid-tropospheric African easterly jet (AEJ). Burpee (1972) showed that the climatological basic state over North Africa is associated with a reversal in the meridional gradient of potential vorticity (PV), satisfying the necessary condition for mixed baroclinic-barotropic instability (Charney and Stern, 1962). He proposed that AEWs are the result of small amplitude disturbances growing on the unstable basic state. In this

viewpoint, AEWs amplify at the expense of the background reservoir of zonal kinetic energy and zonal available potential energy (e.g., Hsieh and Cook, 2005). Calculations made using field campaign data have generally supported this notion (e.g., Norquist et al., 1977; Reed et al., 1977). Furthermore, the fastest growing (or slowest decaying) normal modes of the AEJ in idealized numerical models appear to have wavenumber and frequency that are close to observed AEWs (e.g., Rennick, 1976; Simmons, 1977; Thorncroft and Hoskins, 1994; Leroux and Hall, 2009). In this regard, hydrodynamic instability has been a

useful model to account for the existence of AEWs.





## 1.1 Criticisms of linear normal mode instability

In the recent past, the applicability of linear normal mode instability to AEWs has been questioned. Two main criticisms have been put forth. The first one concerns the impact of viscous damping. Hall et al. (2006) showed that AEW growth rates were reduced, or even reversed, by what they take to be realistic levels of damping in an idealized general circulation model. They

contended that an AEJ, that is otherwise super-critical to inviscid normal modes, may be stabilized by boundary layer damping. The second criticism concerns the localized nature of the AEJ. Some studies have claimed that the zonal extent of the AEJ is too short to sustain wave growth. Thorncroft et al. (2008) estimated that the AEJ length is no more than twice the wavelength of an AEW. They claimed that the limited extent of the AEJ would preclude unstable modes from emerging spontaneously out of background noise. This assertion was repeated by Leroux and Hall (2009). These two lines of argument led Thorncroft et al.

(2008) to suggest that large amplitude triggers are necessary for the generation of AEWs.

Whereas Hall et al. (2006) and Thorncroft et al. (2008) used a climatological basic state, Leroux and Hall (2009) used the same model but with 336 different states derived from global reanalysis data. While many of their basic states showed no development, some showed growing waves. They reiterated the conclusion of Thorncroft et al. (2008) regarding the importance of external triggers for AEWs. They also concluded the growth rate of waves was most consistently related, not necessarily

to the strength of the AEJ, but to the vertical wind shear associated with it. In a related study, Leroux et al. (2010) showed that external triggers from extratropics can also account for AEW activity. The triggering hypothesis for AEWs is now widely accepted view of AEW formation. Yet, it should be noted that there is little doubt that observed and modeled waves appear to be sustained by baroclinic and barotropic energy conversions from the background state of the atmosphere over North Africa (e.g., Norquist et al., 1977; Thorncroft and Hoskins, 1994; Hsieh and Cook, 2007).

## 1.2 The antifriction role of moist convection and dust aerosol forcing

While the results of Hall et al. (2006) and Leroux and Hall (2009) have spurred the adoption of the viewpoint that AEWs need an external trigger, at least two critical aspects of the dynamics are missing in their simulations. Their model had no interactive moist convection. The AEW stormtrack region of North Africa is home to frequent and intense mesoscale convective systems (e.g., Laing et al., 1999; Fink and Reiner, 2003; Laing et al., 2011). It has been shown that moist convection is essential to

account for observed AEW structure and strength (e.g.. Mass, 1979; Schwendike and Jones, 2010; Hsieh and Cook, 2008; Mekonnen and Rossow, 2011; Berry and Thorncroft, 2012b; Janiga and Thorncroft, 2014; Tomassini et al., 2017; Russell et al., 2019). Both time-mean and transient moist convection act to destabilize AEWs (Russell and Aiyyer, 2020). Although it can be argued that the basic states used by Hall et al. (2006) and Leroux and Hall (2009) have some imprint of time-mean convection, the lack of wave-coupled moist convection severely underestimates their potential growth rate.

AEWs are also subject to strong aerosol radiative forcing associated with the Saharan mineral dust (SMD) (e.g., Karyampudi and Carlson, 1988). There is increasing evidence that SMD can lead to AEW amplification (Jones et al., 2004; Ma et al., 2012). Grogan et al. (2016) showed that the radiative forcing associated with SMD can enhance the eddy available potential energy (EAPE) such that AEW growth rates can be significantly amplified relative to dust-free conditions. Using analytical solutions





and model simulations, Nathan et al. (2017) showed that even a sub-critical AEJ, wherein the background PV gradient is
single-signed, can yield AEWs that are destabilized by the SMD. They showed that the radiative-dynamical feedback due to
SMD can offset the low-level damping.

Taken together, it can be argued that the antifriction behaviour of moist convection and dust radiative forcing should be an
important consideration while addressing the first of the two criticism of linear normal mode instability. This does not imply
that finite amplitude triggers have no role to play. Instead, it simply means that they are not necessary as claimed by Hall et al.
(2006). It is certainly plausible that disturbances induced by large convective outbreaks and extratropical intrusions can project
on the normal modes, yielding weak, but finite amplitude waves that can then exponentially grow via dust-radiative instability
and moist convection. This will crucially depend on how much the external forcing projects on the scale of AEWs. Indeed,
Thorncroft et al. (2008) found that the waves in their model were much weaker than observations even though the amplitude of
the forcing was of a reasonable scale, representing the action of multiple mesoscale convective systems. They also recognized
that coupling with moist convection was needed to account for observed wave amplitudes.

## 2   Objective and background

As noted earlier, in the recent past, two arguments against linear normal mode instability have been put forth. The first one
regarding the stabilizing effect of frictional damping been addressed in several studies, as discussed above, by including the
effects of moist convection and dust radiative radiative forcing. The second criticism regarding the limited zonal extent of the
instability is yet to be convincingly examined. Our objective is to address it by elucidating a key property of the wavepackets
generated in the localized AEJ.

### 2.1   AEW Wavepackets

Past studies have established that AEW wavelength ($\lambda$), and westward phase speed ($c_p$) are about 3000 km and 10 m$s^{-1}$
respectively. Estimating the zonal extent ($L$) of the AEJ to be around $2\lambda$, an approximate residence time is:

$$\tau_p \approx \frac{L}{c_p} \approx \frac{2\lambda}{c_p} \approx 7 \text{ days} \qquad (1)$$

The argument that the AEJ is too localized (e.g., Thorncroft et al., 2008; Leroux and Hall, 2009) essentially is the statement that
a week's time is not sufficient for normal modes to emerge out of background noise. However, an important consideration that
is missing from this viewpoint is that the reference speed here should reflect the group propagation instead of the phase. This
notion is not new. It has been demonstrated convincingly that extratropical baroclinic eddies are best modeled as modulated
wavepackets and their dynamics are intimately tied to their group propagation (e.g., Pierrehumbert, 1984; Mak and Cai, 1989;
Orlanski and Katzfey, 1991; Chang et al., 2002; Swanson, 2007). Following this lead, Diaz and Aiyyer (2013a) and Diaz and
Aiyyer (2013b) showed that composite AEWs in global reanalysis fields appeared as dispersive wavepackets.

Figure 1 shows time-longitude slices of eddy kinetic energy (EKE) at two different levels. The left panel shows 650 hPa EKE
averaged over 5–15 °N for 2006. The right panel shows 925 hPa EKE averaged over 15–25 °N for 2008. EKE is calculated

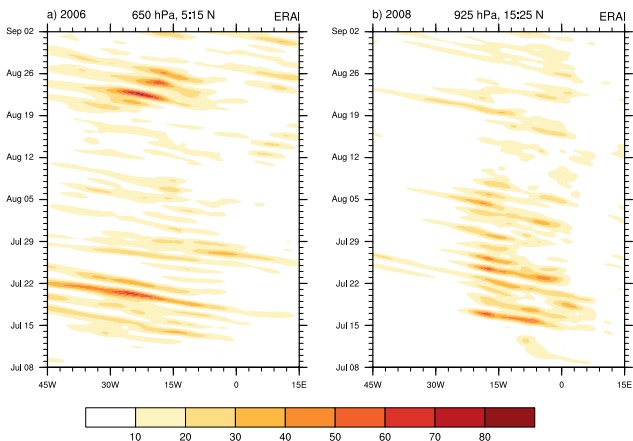

**Figure 1.** Time-longitude diagram of eddy kinetic energy (J kg$^{-1}$) from ERAi for July-August for the year 2006 at 650 hPa, averaged over 5 to 15 °N (a) and 2008 at 925 hPa, averaged over 15 to 25 °N (b).

using 2-10 day band-pass filtered winds from the European Centre for Medium-Range Weather Forecasts (ECMWF) interim reanalysis (ERAi). These two levels highlight the northern and southern AEW stormtracks. Both panels of Fig. 1 show distinct wavepackets that appear to disperse upstream and downstream. The key observations are (i) the wavepackets are co-located with the AEJ, which extends from about 30$^o$W–30$^o$E, and (ii) the group speed is significantly smaller than the phase speed. With the group speed as the reference propagation metric, we can estimate the residence time as:

$$\tau_g \approx \frac{L}{c_g} \approx \frac{2\lambda}{c_g} \tag{2}$$

and with $c_g << c_p$, we have

$$\tau_g >> \tau_p \tag{3}$$

This states that the residence time is mediated by the group propagation dynamic. In the case of a slowly propagating wavepacket, this indicates that even a localized AEJ can support growth by the combined effects of the jet instability, moist convection and SMD radiative forcing. In subsequent sections, we explore the structure of the AEW wavepacket's group speed and the implication for its instability.

## 2.2 Spatial Instability

Using a local kinetic energy budget, Diaz and Aiyyer (2013a) showed that upstream energy dispersion allowed new development within the lagging edge of a slowly propagating AEW wavepacket. Their results also suggested the intriguing potential for the AEJ to support absolute instability, a form of spatial hydrodynamic instability (e.g., Pierrehumbert, 1984; Huerre and Monkewitz, 1990). Herein, both temporal and spatial growth play a role in the development of disturbances. The instability can be classified as either *absolute* or *convective*, with the delineation arising due to two possibilities when a disturbance is introduced within an unstable plane-parallel jet (Briggs, 1964; Pierrehumbert, 1984; Dunkerton, 1993).




- A developing wavepacket grows and disperses upstream and downstream of its excitation point. Any fixed point in the domain eventually experiences exponential growth. This represents absolute instability and a conceptual illustration for easterly flow in shown in Figure 2a.

- A developing wavepacket continues to grow but is unable to spread sufficiently upstream and is swept away by the flow such that a fixed point sees growth followed by decay in wave amplitude. This represents convective instability and a conceptual illustration for easterly flow in shown in Figure 2b.

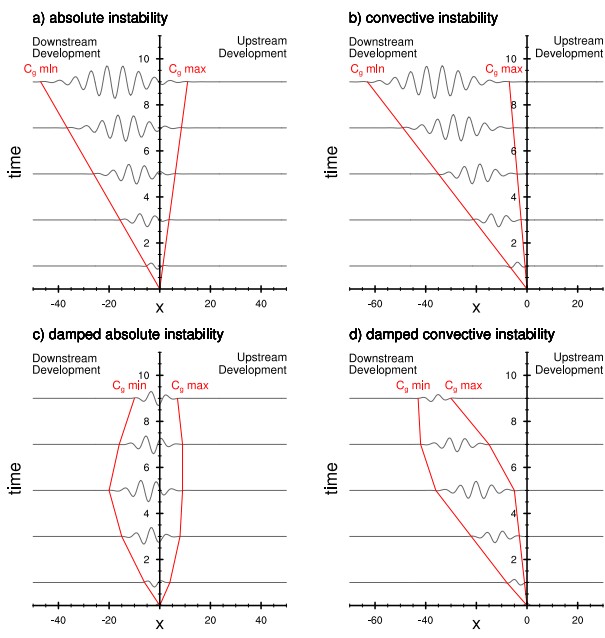

**Figure 2.** Absolute (a,c) and convective (b,d) instability conceptually represented in the $x$-$t$ plane for an unstable easterly flow. If $C_g = 0$ within the cone, then there is both upstream *and* downstream development, and the flow is absolutely unstable (a). Otherwise, the flow is convectively unstable (b). If the growth of the wavepacket is sufficiently damped as in (c) and (d), the amplitude of the wavepacket will eventually decay to zero, resulting in weaker waves and a shorter lived wavepacket. Adapted from Diaz and Aiyyer (2015).

When the region of instability is zonally confined, a wavepacket that is unable to disperse upstream will cease to grow after it exits this region. On the other hand, with both upstream and downstream dispersion, an absolutely unstable wavepacket can continue to grow at all subsequent times. If the basic state cannot support *any* development, it is deemed stable. It is also possible that a basic state that is unstable under inviscid conditions can be stabilized by damping (e.g., Hall and Sardeshmukh, 1998). In the atmosphere, sources of damping include boundary layer friction, radiative and convective damping. Thus, in the presence of damping, the growth rate from either of the two spatial instabilities can be reduced or reversed. This is shown in the conceptual diagrams in Figures 2c,d. We will refer to these situations as damped absolute and convective instabilities respectively.





Diaz and Aiyyer (2015) examined the stability of zonally uniform version of the climatological AEJ using direct numerical simulations. In their model, a stationary spreading and growing wavepacket persisted for several days, even with moderate

parameterized damping. They estimated the total group velocity ($C_g$) following the method described in Orlanski and Chang (1993). In the lieu of a rigorous analytical determination of spatial instability, the heuristic approach (e.g., Orlanski and Chang, 1993) as adapted for easterly waves by Diaz and Aiyyer (2015) is useful:

$$\left(C_g\right)_{\min} < 0 < \left(C_g\right)_{\max} \Rightarrow \text{absolute instability, and}$$

$$\left(C_g\right)_{\min} < \left(C_g\right)_{\max} < 0 \Rightarrow \text{convective instability.} \tag{4}$$

Based on the analysis of their simulations, Diaz and Aiyyer (2015) claimed that a zonally uniform representation of the AEJ was absolutely unstable. Given that moist convection and dust radiative effects can be additional sources of destabilization, this suggests that absolute instability may be a viable mechanism for the existence of AEWs.

## 2.3 Hypotheses

Several aspects of the results of Diaz and Aiyyer (2015) ought to be explored further. First, their AEJ did not capture the
streamwise inhomogeneity of the observed jet. Second, they only considered a single climatological basic state. They also speculated that the basic states used in Thorncroft et al. (2008), Hall et al. (2006), and Leroux and Hall (2009) were convectively unstable. This is yet to be confirmed. Additionally, Fig. 1 also highlights the intermittent AEW activity. Since AEWs tend to propagate in groups, this yields alternate periods of enhanced and reduced synoptic activity. For example during 2006, two distinct wavepackets can be seen starting around July 14 and August 19, with maximum activity just west of -15°E. Although
Fig. 1 shows only two selected examples, such interspersed episodes of enhanced AEW activity are commonly observed each year. It is of interest to identify factors that mediate this episodic nature of AEW wavepackets. Based on the preceding discussions, we identify two questions and attendant hypotheses to address our objective.

1. How do persistent AEW wavepackets develop in the zonally localized AEJ?

   *Hypothesis*: AEW persistence is related to the generation of a nearly stationary wavepacket within a realistic, localized
AEJ. This specifically addresses the concern raised in previous studies (e.g., Thorncroft et al., 2008; Leroux and Hall, 2009).

   We note that one of the hallmarks of absolute instability is a wavepacket that disperses upstream and downstream, with a zero group speed somewhere in the packet. However, a formal determination of absolute instability requires the investigation of the dispersion relationship in the complex plane (e.g., Pierrehumbert, 1984; Dunkerton, 1993). Since that
is beyond the scope of this effort, we do not make a formal claim of absolute instability. As in Diaz and Aiyyer (2015), but with a fully varying background flow, we only present a heuristic analysis based on direct numerical simulations.

2. What mediates the vacillation of the episodes?

   *Hypothesis*: A slowly varying background flow alternates aperiodically between supporting stationary and traveling wavepackets, leading to the AEW episodes.





In the context of spatial instability, this posits that AEW episodes are mediated by transition between the nature of the instability of the background state (i.e., absolute → convective; and convective → absolute). This also suggests the possibility that intermittent AEW activity may be related to an internal mechanism involving the instability of the AEJ in addition to previously suggested external sources of forcing.

    It should be noted that issue of variability of AEW activity mediated by external mechanisms has been examined in several

past studies. Leroux et al. (2011) determined that extratropical disturbances from the North Atlantic stormtrack can influence AEW activity. Others have documented the impact of equatorial Kelvin waves and the Madden Julian oscillation on the intraseasonal modulation of AEWs (e.g. Matthews, 2004; Leroux and Hall, 2009; Ventrice et al., 2011; Alaka and Maloney, 2012, 2014). While moist convection has been shown to be important to evolution of AEWs, in the spirit of retaining only essence of the dynamics, we do not explicitly account for its feedback in our approach. We test these hypotheses by employing

an idealized general circulation model.

## 3   Primitive Equation Model

We use the the University of Reading multilevel primitive equation model configured as described in Hall et al. (2006). We choose this model for several reasons. It is the same model that was used in related studies discussed earlier (Thorncroft et al., 2008; Leroux and Hall, 2009; Leroux et al., 2011). By using a consistent model and experimental approach, we can not only

build upon their work but also provide clarification and novel interpretation of their results. We provide below some information on the model configuration for completeness.

    The model has a horizontal spectral resolution of T31 for 10 equally spaced sigma levels. The full nonlinear equations for tendencies of temperature, vorticity, divergence and $\log$(surface pressure) are integrated using a semi-implicit time step of 22.5 minutes. A $\nabla^6$ diffusion is implemented for the momentum and temperature fields. As in Hall et al. (2006), low-level damping

is also applied to the momentum and temperature fields, intended as a modest representation of near-surface turbulent heat and momentum transfer. The damping rates in the lower levels linearly decrease from the surface over $0.8 \leq \sigma \leq 1.0$ with an average e-folding time scale of 2 days for momentum and 4 days for temperature. In the free atmosphere ($\sigma < 0.8$), damping time scales are 10 days for momentum and 30 days for temperature. In addition, we also damp the areas of the globe poleward of $30°$ in both hemisphere to preclude development in the extratropical stormtracks.

Since the model equations are not linearized about a fixed state, a standard approach to maintaining a time-invariant basic state was implemented by Hall et al. (2006). Herein, a forcing term is added to the equations that collectively represent the effects of diabatic heating and transient effects needed to maintain the basic state. The data for the basic state are taken from the National Center for Environmental Prediction/Department of Energy (NCEP/DOE AMIP II) Reanalysis (Kanamitsu et al., 2002).





## 3.1 Wave forcing

To excite waves in the AEJ, we force the model with a pulse of localized heating following the method of Thorncroft et al. (2008). The heat source ($H$) is placed at 15 °N, 20 °E and is switched off after 24 hours into the simulation. This heating is meant to represent the latent heat release associated with several MCSs, and is defined as

$$
H = \begin{cases} H_0 \cos^2\left(\dfrac{\pi}{2}\dfrac{r}{r_0}\right) & r \le r_0 \\ 0 & r > r_0, \end{cases}
\tag{5}
$$

where $r$ is the horizontal radius and $r_0$ is the horizontal bounding radius of the heating with a distance of 5° longitude and latitude. $H_0$ is the *deep convective* profile from Thorncroft et al. (2008) defined as:

$$
H_0 = \frac{\pi}{2}\sin(\pi\sigma).
\tag{6}
$$

The vertically integrated heating rate for this heating is 5 K day$^{-1}$, corresponding to a peak rain rate of 20 mm day$^{-1}$. The initial heating is scaled by a factor of $10^{-4}$ to ensure consistency with linearization about a fixed basic state. The resulting perturbations are later scaled up by the same factor for display.

To reiterate, we use the same method of initiating waves in a fixed basic state as was done by Thorncroft et al. (2008) and Leroux and Hall (2009) to specifically address the concerns raised by them. Our conclusions are not sensitive to the choice of a wave maker in the model. We have forced the model with red noise and other localized perturbations (e.g., finite amplitude Gaussian vortex) and reach the same conclusion as presented in subsequent sections.

## 3.2 Simulations

All simulations span 100 days. To quantify wave amplitude, we use the standard definition of eddy kinetic energy (EKE):

$$
K_e(x,y,\sigma,t) = \frac{1}{2}\mathbf{v}\cdot\mathbf{v}
\tag{7}
$$

where $\mathbf{v}$ represents the perturbation velocity field. We define a quantity $\beta$ as the ratio:

$$
\beta(x,\sigma,t) = \log_{10}\frac{[K_e]}{[K_{e_0}]}
\tag{8}
$$

where the square brackets denote averaging over over 5–20 °N and $K_{e_0}$ is the maximum EKE within the first 24 hours. $\beta$ is not a growth rate but rather a convenient measure of the strength of the wavepacket at any given time relative to the energy input by the initial transient thermal forcing. We use it to classify the longevity of wavepackets. If $\beta < -1$ at the lagging edge within the first 20 days, it is classified as short-lived. If $\beta > -1$ for longer than 20 days but less than the full 100 days of the simulation, it is classified as intermediate-lived, and if it $\beta > -1$ for the full simulation, it is classified as long-lived. This method of classification does not require that the fastest growing or slowest decaying normal mode to be isolated within the simulation period. We examine the evolution of wavepackets in the following 779 basic state configurations:

- climatological basic state: June-September mean over 1987–2017.





– individual basic states: These were created using 15-day averages, with 5 day-overlap, for the same period as above. This yields 775 distinct basic states. Within each year, they represent slow and continuously evolving background flow. This approach of using a sequence of basic states taken from the reanalysis data is similar to Leroux and Hall (2009) who examined a series of basic states taken from global reanalysis. However, their focus was on global temporal growth rates of waves, and they did not perform any spatial diagnostics of the AEW wavepackets.

– ensemble averaged basic states: Out of the 775 individual basic states, subsets based on wavepacket longevity are averaged to create 3 additional basic states. These correspond to the ensemble-averaged short, intermediate, and long-lived basic states.

## 4   Results

Each simulation results in one distinct wavepacket. Of those, 521 (67%) are short-lived, 135 (17%) are intermediate-lived, and 116 (15%) are long lived (Table 1). For brevity, we focus on the results from the climatological and ensemble-averaged basic states.

|  | Number | Percentage |
|---|---|---|
| **Short-lived** | 521 | 67 |
| **Intermediate-lived** | 135 | 17 |
| **Long-lived** | 116 | 15 |

**Table 1.** Categories of wavepackets from the 775 simulations.

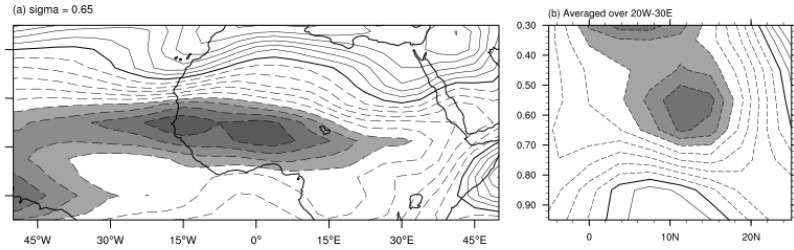

**Figure 3.** Climatological zonal winds ($\bar{U} < -6$ m s$^{-1}$ shaded and negative values dashed) for: (a) $\sigma = 0.65$ level and (b) latitude-height cross section averaged over -15 to 15 °E.



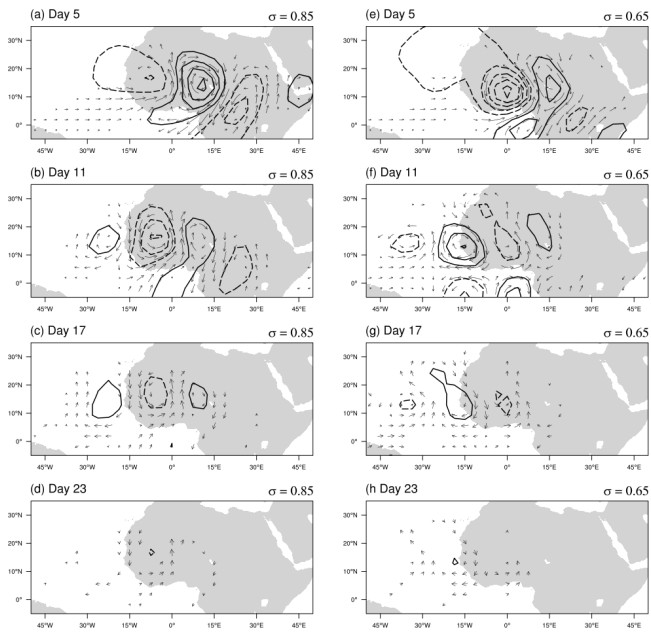

**Figure 4.** Streamfunction (interval $1 \times 10^5$ m$^2$ s$^{-1}$ with negative values dashed) and wind vectors at $\sigma = 0.85$ (left) and $0.65$ (right) showing the wave response within the climatological composite basic state for days 5, 11, 17, and 23 from top to bottom.

## 4.1 Climatological Basic State


We begin with the response to transient heating imposed on the climatological basic state. Figure 3 shows the basic state zonal winds at $\sigma = 0.65$ and a latitude-height cross-section averaged over 15°W–15 °E. The panels show the zonally localized AEJ peaking around 12-15 °N and around 600 hPa along the vertical. The monsoon westerlies are located near the surface. In the first two days of the simulation, the fixed-heating produces a baroclinic vortex. Subsequently, consistent with Thorncroft et al.

(2008), a series of perturbations resembling observed AEWs emerges. Figure 4 shows the resulting perturbation streamfunction at $\sigma = 0.85 (\approx 850$ hPa) and $0.65 (\approx 650$ hPa) on days 5, 11, 17, and 23. The individual phases propagate westward, but the wavepacket remains nearly stationary and slowly decays. Figure 5a shows a Hovmoller diagram of the EKE, averaged over 5–20 °N during the first 40 days of this simulation. The EKE falls below 10% of its initial value by day 20. Therefore, we classify the resulting wavepacket within the climatological basic state as short-lived.

Figure 5b illustrates the parameter $\beta$ calculated using Eq. 8 for the full duration of the simulation. It shows clearly that even though the wavepacket was decaying, it was not swept out of the region, and its structure persisted near 0°E. This suggests that the ground-relative group speed was zero within the wavepacket, which is one of the conditions associated with absolute instability (Equation 4). Importantly, as often seen in observations, the AEW wavepacket is stationary within the AEJ over West Africa. Calculations of energy budget confirm that barotropic and baroclinic energy conversions from the basic state are both





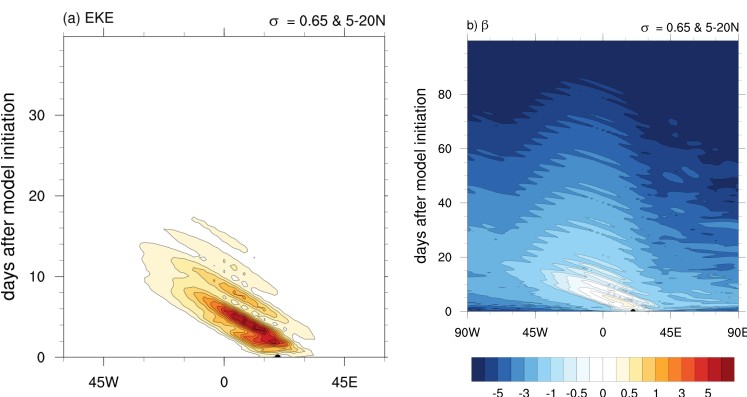

**Figure 5.** Time-longitude diagram of EKE (a, J kg$^{-1}$) and $\beta$ (b) at $\sigma = 0.65$ resulting from fixed-heating on the JJAS 1987-2017 composite basic state averaged over 5–20 °N. Note the difference in times shown between (a) and (b). The black dot represents the location of the initial heating perturbation.

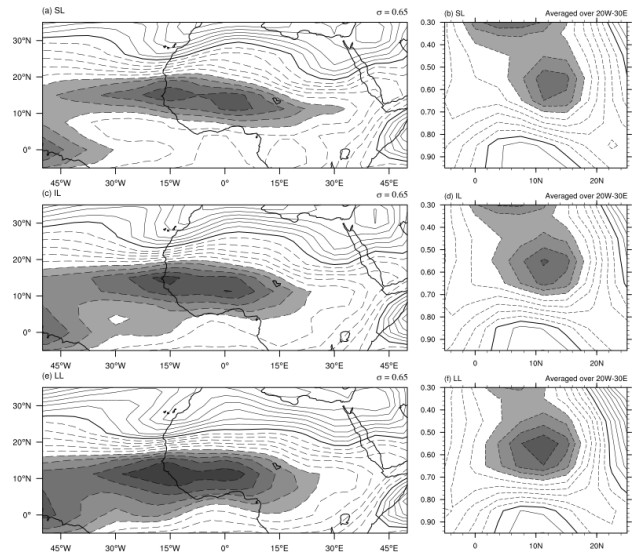

**Figure 6.** Zonal winds ($< -6$ m s$^{-1}$ shaded with negative values dashed) at $\sigma = 0.65$ (left column) and latitude-height cross section averaged over -15 to 15 °E (right column) for the short-, intermediate-, and long-lived basic states.

sources of EKE (not shown). However, the presence of damping in the model leads to the eventual decay of the wavepacket. This is consistent with the picture of damped absolute instability (Figure 2c).

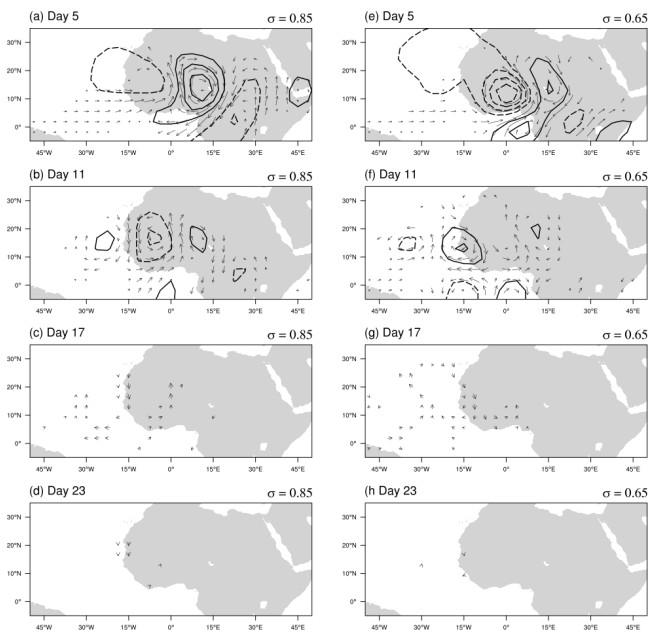

**Figure 7.** Streamfunction (interval $1 \times 10^5$ m$^2$ s$^{-1}$ with negative values dashed) and wind vectors at $\sigma = 0.85$ (left) and 0.65 (right) showing the wave response within the short-lived basic state for days 5, 11, 17, and 23 from top to bottom.

## 4.2 Wave responses for the ensemble averaged categories

We now consider the ensemble-average basic states for the short, intermediate and long-lived wavepackets. The zonal winds at $\sigma = 0.65$ and a height-latitude cross section for the three basic states are shown in Figure 6. The basic-state AEJ gets

progressively stronger and shifts equatorward moving from short to long lived wavepackets. The jet in the former reaches a peak of about 9 m s$^{-1}$ and in the latter about 10 m s$^{-1}$. As the AEJ is more directly above the surface westerlies and stronger in the long-lived basic state, both horizontal and vertical wind shear are enhanced as compared to the other two basic states. Therefore, we anticipate that both barotropic and baroclinic conversion rates will be highest in this basic state. Each of these basic states yield quite similar reversals in the meridional PV gradient (not shown) which are capable of supporting the

development of AEWs through mixed barotropic-baroclinic instability. Consistent with it, energy budgets confirm barotropic and baroclinic energy conversions from the basic state (not shown).

As in the climatological case, each ensemble-average basic state generates a series of disturbances that resemble AEW packets. Figure 7 shows short-lived response for days 5–23, separated by roughly one wave-period. On day 5, the streamfunction field shows waves with horizontal tilt consistent with barotropic instability. The wavepacket has its peak amplitude around 0 °E.

On day 11, the amplitude has dropped substantially, and by day 23, the streamfunction fails to reach the minimum contour used in the figure. Figure 8 shows the evolution of the intermediate-lived wavepacket. The streamfunction on day 5 closely resem-



bles the same for the short-lived wavepacket but with greater amplitude. This suggests that the basic state for intermediate-lived wavepacket permits larger growth rates. By day 11, the wavepacket is still present but has undergone some damping, although not nearly to the same extent of the short-lived wavepacket. On day 17, the wavepacket persists but is weaker. By day 23, the

AEW packet has further diminished. Figure 9 shows shows the evolution of the long-lived wavepacket. As opposed to the short and intermediate lived wavepackets, this one continues to grow at all subsequent times after initiation. The waves are tilted upshear in both the horizontal and vertical planes, indicating barotropic and baroclinic energy conversions.

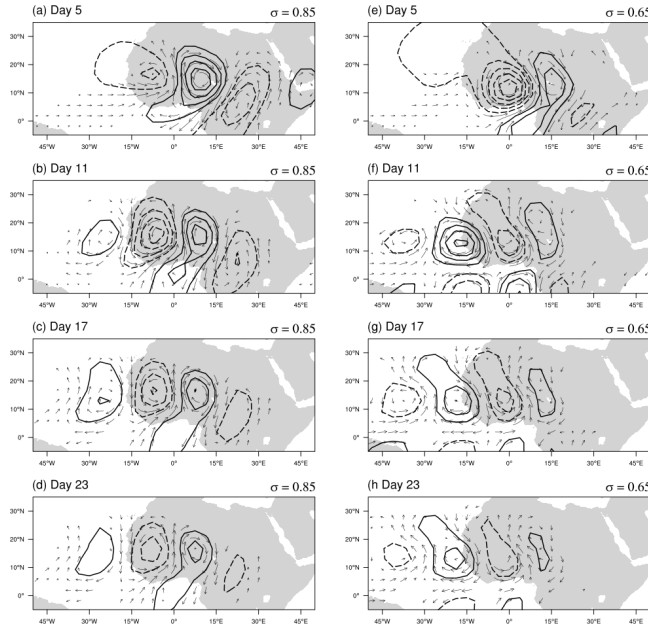

**Figure 8.** Streamfunction (interval $1 \times 10^5$ m$^2$ s$^{-1}$ with negative values dashed) and wind vectors at $\sigma = 0.85$ (left) and 0.65 (right) showing the wave response within the intermediate lived basic state for days 5, 11, 17, and 23 from top to bottom.

Hovmoller diagrams for EKE and $\beta$ for these three simulations are shown in Figure 10. The former is shown for the first 30 days while the latter for the full 100 days of the simulation. The location of the center of the transient heating is shown

using a black dot. The EKE panels for the short and intermediate-lived cases show damped wavepackets while the long-lived panel shows an amplifying wavepacket. Owing to the logarithmic definition of $\beta$, we can clearly see the behavior of the wavepackets long after the transient heating. Initially, the response is a baroclinic vortex that projects on a wide range of modes. Some of those are advected out of the localized region of instability and damped. In the short-lived case, the wavepacket that remains is much weaker and is not sustained by barotropic-baroclinic energy conversions against the imposed frictional

damping. This is illustrated by the progressively negative values of $\beta$ within the vicinity of 0 °E. After 60 days, the wavepacket amplitude is indistinguishable from noise in the model. In the intermediate-lived simulation, the damping does not diminish the wavepacket as quickly. As a result, the wavepacket is visible throughout its lifetime. However, the EKE drops below 10% of its



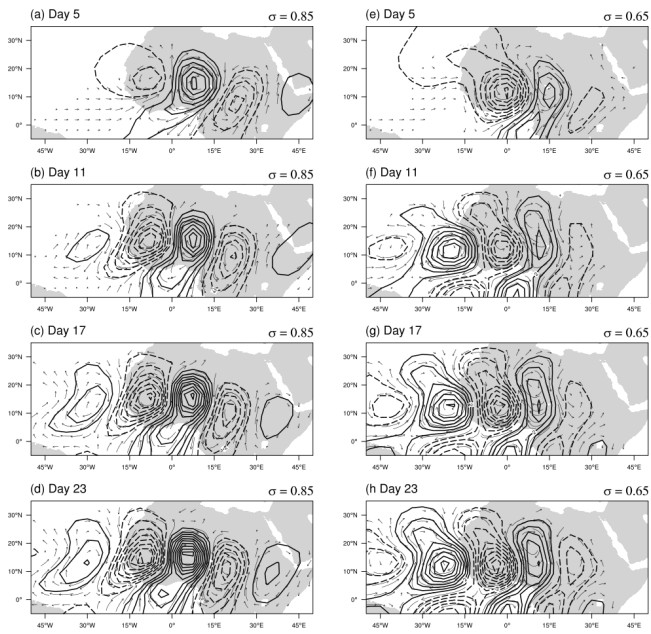

**Figure 9.** Streamfunction (interval $1 \times 10^5$ m$^2$ s$^{-1}$ with negative values dashed) and wind vectors at $\sigma = 0.85$ (left) and 0.65 (right) showing the wave response within the long-lived basic state for days 5, 11, 17, and 23 from top to bottom.

original value by day 40. The long-lived case clearly shows a growing, expanding wavepacket. By the end of the simulation, its amplitude reaches two orders of magnitude higher than its initial value.

Another illustration of the behavior of the wavepackets in the three simulations is presented in Figure 11. Here, the stream-function ($\sigma = 0.65$ level), averaged over 10–20°N, is plotted against longitude for selected times. Note the different ranges for the ordinate in the three panels. Taken together with Figure 10, it is evident that the wavepacket in all three simulations remains remains nearly stationary. In the short and intermediate-lived cases, the leading and lagging edges of the AEW packet slowly seem to be collapsing towards 0 °E. They resemble the conceptual diagram for damped absolute instability shown in Figure 2c.

On the other hand, the amplification and expansion of the wavepacket in the long-lived case resembles the conceptual diagram for absolute instability shown in Figure 2a.

Figure 12 shows the the global averaged EKE and its growth rate, with the latter defined as:

$$\frac{1}{K_e} \frac{\partial K_e}{\partial t}$$

In each simulation, the transient response to the fixed-heating forcing causes an initial spike in EKE, followed by a period of

adjustment after the heating is switched off. As expected, the short and intermediate-lived wavepacket growth rates are negative owing to the stabilization effect of the imposed friction. The long-lived case shows exponential growth. It appears that 20 days



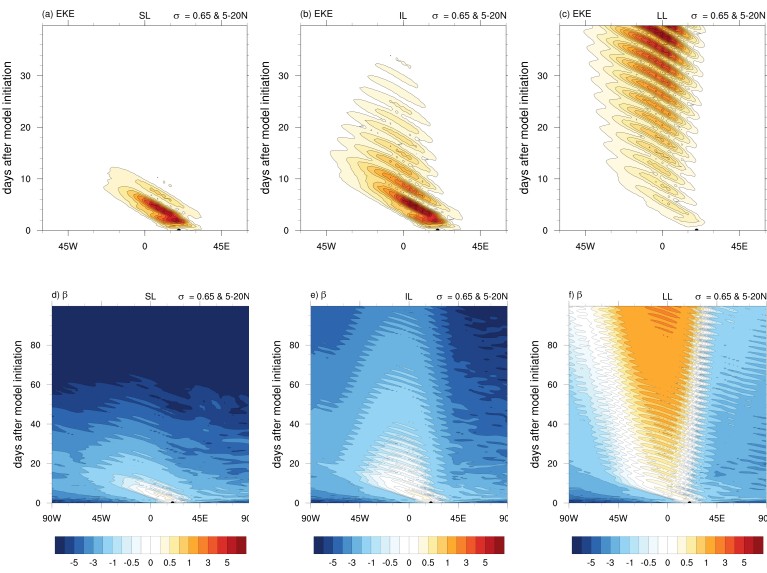

**Figure 10.** Time-longitude diagram of EKE (top) and $\beta$ (bottom) at $\sigma = 0.65$ resulting from fixed-heating on the short- (a,d), intermediate-(b,e), and long-lived (c,f) composite basic states averaged over 5–20 °N. The black dot represents the location of the initial heating perturbation.

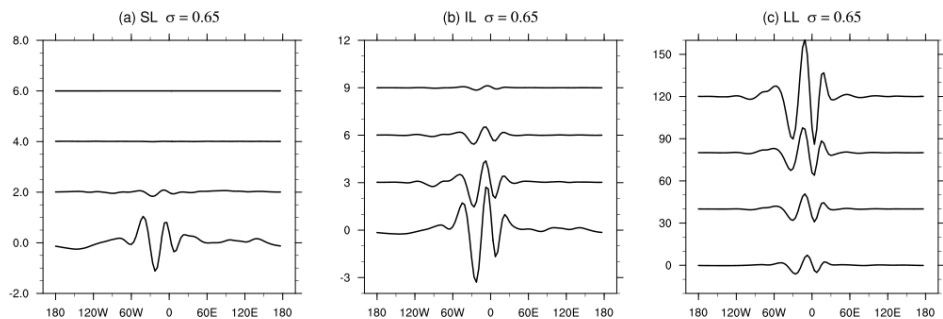

**Figure 11.** Streamfunction ($10^5$ m$^2$ s$^{-1}$) averaged over 10–20 °N at $\sigma = 0.65$ and plotted as a function of longitude for selected days for short- (a), intermediate- (b), and long-lived (c) simulations. Each day's streamfunction is offset along the ordinate for visualization.

of simulations is sufficient to reach this steady growth rate and signifies the emergence of the dominant normal mode for the localized jet.

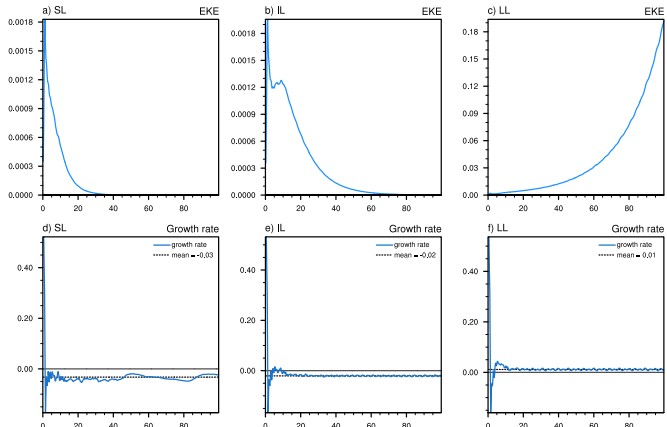

**Figure 12.** Time-series of the globally averaged EKE (top, J kg$^{-1}$) and growth rates (bottom, day$^{-1}$) resulting from the fixed-heating anomaly on the short- (a,d), intermediate- (b,e), and long-lived (c,f) basic states. The mean growth rate, averaged over the last 50 days of the simulation, is shown by the dashed line.

## 4.3 Group speed

Diaz and Aiyyer (2015) adapted the method of Orlanski and Chang (1993) to calculate the group speed across their simulated wavepackets. We write it for the zonal group speed as follows:

$$C_g = \frac{\iint (u_a \Phi' + \mathrm{T}_e \bar{U}) \, \mathrm{d}p \, \mathrm{d}A}{\iint \mathrm{T}_e \, \mathrm{d}p \, \mathrm{d}A}, \tag{9}$$

where $u_a$ is the perturbation zonal ageostrophic wind, $\Phi'$ is the perturbation geopotential, $\bar{U}$ is the basic state zonal wind, and $\mathrm{T}_e$ is the total eddy energy, which was defined by Orlanski and Katzfey (1991) as the sum of the eddy kinetic energy and the 295 eddy available potential energy:

$$\mathrm{T}_e = \frac{1}{2}\mathbf{v} \cdot \mathbf{v} - \left( \frac{\alpha_m}{2\Theta_m} \frac{\theta^2}{d\tilde{\Theta}/dp} \right), \tag{10}$$

where $\mathbf{v}$ is the perturbation velocity , $\alpha_m$ is the time mean specific density, $\Theta$ is the potential temperature, $\tilde{\Theta}$ is its horizontal average and $\theta$ is the perturbation potential temperature. As written above, $C_g$ includes the contribution of both the ageostrophic geopotential flux and the energy transport by the background flow. To apply Equation 9 to our model output, we integrate over 300 latitudes 5°S –30°N in the meridional direction, half-wavelengths in the zonal direction, and the entire depth of the model in the vertical direction. The group speed is further averaged over days 30–35 for display. Figure 13 shows the calculation for the climatological and the three ensemble-averaged basic states. In all four cases, the group speed is westward on the leading edge, eastward on the lagging edge, and vanishes somewhere in between. For the short-lived case this happens at 30 °W, for intermediate-lived around 10 °W, and the long-lived around 0 °E. Owing to upstream and downstream dispersion, portions of 305 the wavepacket remain within the region of instability. Thus, the condition for inviscid absolute instability for easterly mean

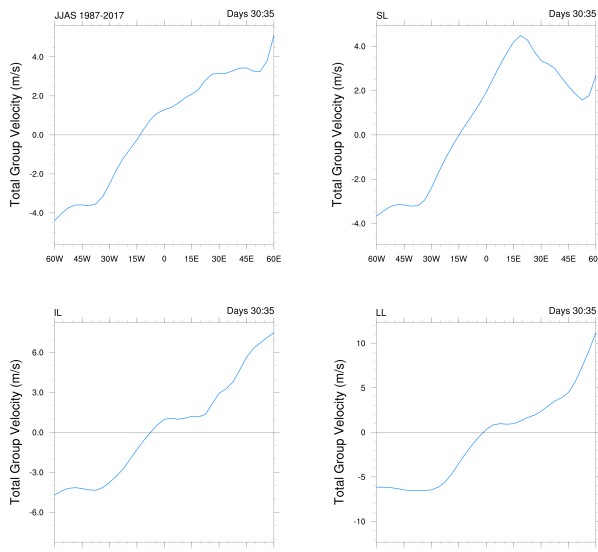

**Figure 13.** Total group velocity (m s$^{-1}$) averaged over days 30–35 of the simulation for the four basic states as marked on the panel: The JJAS long-term climatology, short-lived (SL), (a), intermediate-lived (IL), and long-lived (LL). $C_g$ is evaluated from Equation 9, integrated over half-wavelengths in the zonal direction, from -5 to 30 °N meridionally, throughout the model depth in the vertical direction.

flow (Equation 4) is satisfied in all four ensemble-averaged basic states. The inclusion of damping stabilizes three of them. However, In light of our earlier discussion regarding the destabilizing role of moist convection and SMD, the wavepacket's group dynamic provides a means for a persistent structure and amplification that can overcome damping via coupling with these additional energy sources.

**4.4 Sensitivity to sponge region**

The simulation for the long-lived case shows an exponentially growing stationary wavepacket. Since the model is global, and allows for reentry of waves, it is of interest to examine whether this affects the growth rate. A standard approach to minimizing re-entrant waves is to include a sponge region of damping (e.g., Dunkerton, 1993). We repeat our simulation for the long-lived case by imposing additional damping outside $60^o$W–$60^o$E. The growth rate and the structure of the wavepackets for this simulation are shown in Fig. 14. A comparison with Fig. 11 and Fig. 13 shows that the outcome, with and without the sponge region, are similar. This provides confidence that the wavepacket response and amplification is not dependent on reentering waves. This also points to the possibility of the existence of a mode that is characterized by local absolute instability. This, however, needs to be confirmed more rigorously.





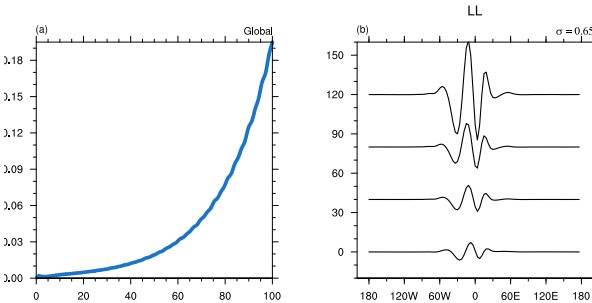

**Figure 14.** Results for the long-lived wavepacket simulation with added sponge region: (a) time-series of the globally averaged EKE (J kg$^{-1}$); and (b) Streamfunction ($10^5$ m$^2$ s$^{-1}$) averaged over 10–20 °N at $\sigma = 0.65$ and plotted as a function of longitude for selected days.

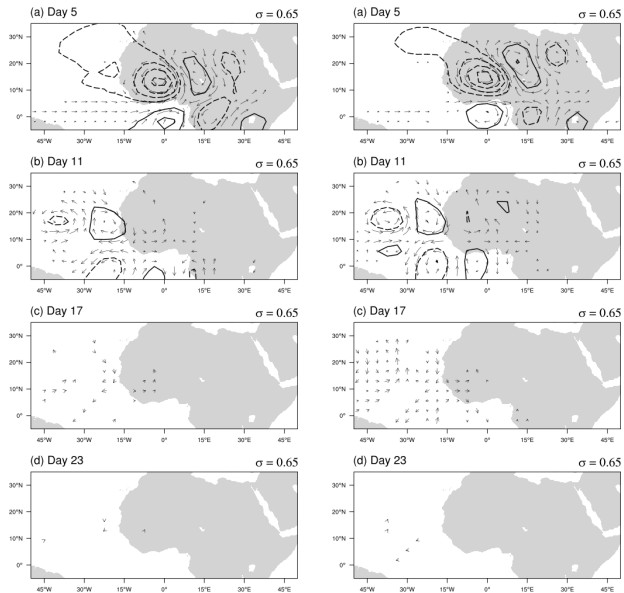

**Figure 15.** Streamfunction (interval $1 \times 10^5$ m$^2$ s$^{-1}$ with negative values dashed) showing the wave response within the basic state formed by the 15-day average centered around August 15, 1995 (left column) and September 4, 2006 (right column).

## 4.5 Examples of Convective Instability

When we examine each of the 775 simulations individually, we find that the majority of them satisfy the criterion for inviscid absolute instability as given in Equation 4. However, most of them are stabilized by damping and fall under the short-lived category. Only 22 cases satisfy the criterion for inviscid convective instability with westward group speed over the entire wavepacket. We now briefly show two examples: one for the basic state taken from the 15 day average centered on August





15, 1995, and the other on September 4, 2006. The perturbation streamfunction ($\sigma = 0.65$ level) for these two simulations

are shown in Figure 15. In both cases, the resulting wavepacket after 5 days looks quite similar to the short-lived composite

case. However, over time, the wavepacket slowly moves beyond the coast of West Africa and is damped. The growth rates for

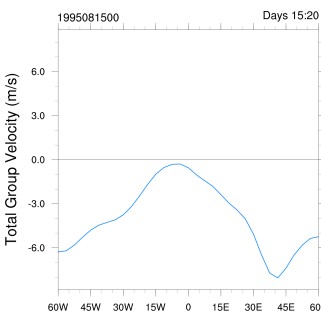

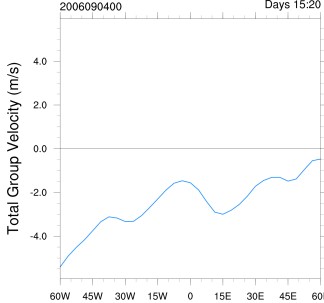

**Figure 16.** Total group velocity (m s$^{-1}$) averaged over days 15–20 of the simulations for the August 15, 1995 basic state (left) and the September 4, 2006 basic state (right).

these two cases are similar to the short-lived basic state simulations (not shown). The Hovmoller diagrams for EKE and the

$\beta$ parameter look similar to the short-lived case (not shown). However the total group velocity (Figure 16) shows uniformly

negative values. This indicates that that the entire wavepacket propagates downstream. Note that we average the group velocity

over days 15–20 in this case. This is because the wavepacket is mostly advected out of the region of instability after this period.

In the absence of damping, the wavepacket would experience growth as long as it remains within the area of instability. Once

it exits the region, it will cease to experience growth. However, as seen in Figure 15, the presence of damping stabilizes the

wavepacket, and it decays after initiation even while passing through the unstable region associated with the AEJ.

## 5   Discussion

The objective of this work is to examine some characteristics of wavepackets generated within the AEJ. In part, this is motivated

by previous studies that have questioned the ability of the zonally localized AEJ to support growing AEWs (e.g. Thorncroft

et al., 2008; Leroux and Hall, 2009). Our work is guided by two hypotheses that we test using an idealized model.

Following the method used by Thorncroft et al. (2008) and Leroux and Hall (2009), we examine the response to transient

heating within the AEW stormtrack using an idealized general circulation model. We begin with a climatological basic state

and then a series of 15-day average states. Out of the 775 simulations, 67% of the wavepackets are short-lived, 17% are

intermediate-lived, and 15% were long-lived. The basic states that result in each of the three categories are further combined to

yield three ensemble-averaged states and three additional simulations are performed. In all three simulations, a nearly stationary

wavepacket ensues and is located over west Africa. The short and intermediate-lived wavepackets are eventually damped but

the long-lived wavepackets remain unstable. The group velocity for these three wavepackets, and indeed most of the basic





states considered here, are found to be westward on leading edge and eastward on the lagging edge of the wavepacket. Only 22 cases showed uniformly westward group velocity across the wavepacket.

All ensemble-averaged basic states considered here are associated with reversal in potential vorticity gradients, and thus satisfy the criterion for inviscid barotropic-baroclinic instability. Consistent with that, the wave structures exhibit upshear tilt in horizontal and vertical planes. As expected, both barotropic and baroclinic conversions of energy from the basic state to
the waves are found (not shown). The scope of the present work does not include a detailed examination of the differences in the basic state to account for the wavepacket behavior. Nonetheless, we note that the long-lived basic state had more vertical and horizontal shear associated with the AEJ as compared to the short-lived basic state. As seen in Figure 6, the AEJ in the long-lived basic state is stronger and is more directly above the surface monsoon westerlies. This yields stronger horizontal and vertical wind shear and consistent with increased barotropic-baroclinic energy conversions. This is reflected in the higher
growth rate of the long-lived wavepacket (Figure 12).

Returning to our motivating hypotheses and questions, our results suggest that the background flow over west Africa supports near-stationary wavepackets. Furthermore, wavepacket diagnostics indicate that the heuristic condition for inviscid absolute instability (e.g., Orlanski and Chang, 1993; Diaz and Aiyyer, 2015) is satisfied. This is the case not only for the climatological state but also for the majority of individual basic states from June-September 1987–2017. Most of our simulations produce only
short-lived AEW packets, indicating that damping can still stabilize an otherwise super-critical jet. Occasionally, however, the amplification via hydrodynamic instability does overcome damping and leads to exponential growth. Importantly, in nearly all cases, the wavepacket remains within the area of instability over west Africa, thereby increasing the potential for interaction with moist convection and SMD that are ubiquitous here.

We used the same experimental method as Thorncroft et al. (2008) and Leroux and Hall (2009) in order to maintain consis-
tency and connections with previous related studies. However, our conclusions are not tied to the mode of wave forcing. Our results do not discount the role of large amplitude triggers of local or remote origin. If these externally forced disturbances project on to the AEW modes, they can lead to subsequent amplification within the AEJ with additional destabilization provided by moist convection and dust radiative feedback. Indeed, Thorncroft et al. (2008) who promoted the idea of external triggers, found that waves in their simulation were much weaker compared to observations even though the external forcing
was of reasonable strength. In the absence of additional destabilization, even triggered waves will fail to amplify. While we have not accounted for interaction with precipitating moist convection and dust aerosol forcing, several other studies have documented their role in destabilizing AEWs (Berry and Thorncroft, 2012a; Schwendike and Jones, 2010; Janiga and Thorncroft, 2014; Poan et al., 2014; Grogan et al., 2016; Tomassini et al., 2017; Nathan et al., 2017; Russell and Aiyyer, 2020; Russell et al., 2019).

If we take the view that the group dynamic of AEWs favors a near-stationary wavepacket within the AEJ, then, the energy conversions associated with the jet, moist convection and dust radiative effects can lead to rapid amplification. The potential for the AEJ to be absolutely unstable raises the possibility that AEW packets can be generated spontaneously within the AEJ and persist for multiple wave periods until nonlinear effects become prominent. The key result is that even though the instability is



zonally localized, the AEW wavepacket is not swept away. This addresses the criticism regarding the limited zonal extent of the AEJ.

Our second hypothesis that intermittent activity of AEWs may be mediated by transition between convective and absolute instability of the basic states is not supported by the results. This is because we find that convective wavepackets are relatively infrequent in our simulations. This may be a limitation of our modeling framework, but nonetheless the implication is that the intermittent nature of AEW activity is likely set externally. Equatorial Kelvin waves and the Madden Julian Oscillation have been shown to modulate AEW activity (e.g. Matthews, 2004; Leroux and Hall, 2009; Ventrice et al., 2011; Alaka and Maloney, 2012, 2014). These external intraseasonal oscillations can modify the background thermodynamic profile and lead to episodes of enhanced and suppressed AEW activity by altering the AEW growth rates.

While our direct numerical simulations suggest the potential applicability of absolute instability, a formal analytical investigation of the spatial instability of the AEJ is warranted. A caveat of our findings is that we use a highly simplified representation of the atmosphere and neglect the feedback with precipitating moist convection and dust radiative feedback on the dynamics of the wavepackets. It is likely that they may play an important role in modifying the nature of spatial instability of AEW packets. This remains to be explored further.

## 6   Conclusion

We find that the background flow over North Africa supports an upstream and downstream dispersing wavepacket that is located within the AEJ. The implication is that, no matter what the source of the initial perturbations may be – spontaneous development or external triggers — the wavepacket is not swept out of the localized region of instability. As a result, AEWs have the opportunity to develop further despite damping, as shown in previous studies, via energy conversions from the jet, and destabilization by moist convection or dust radiative forcing. Our work has shown the importance of the group propagation dynamic for the instability of AEWs.

## 7   Data availability

The reanalysis data used here can be obtained from the European Center for Medium-Range Weather Forecasts ECMWF; https://www.ecmwf.int/en/forecasts/datasets/reanalysis-datasets/era-interim/. The data documentation is provided in Dee et al. (2011). The code for the University of Reading IGCM can be obtained from http://www.met.reading.ac.uk/~mike/dyn_models/igcm/.

## 8   Author contributions

JW conducted the simulations, analysis and produced the visualizations. AA performed some of the analysis and conducted the sponge region experiments. JW and AA wrote computer code for analysis and visualization, and both wrote the text of the paper.





## 9    Competing interests

The authors declare that they have no conflict of interest.

## 10    Financial Support

This work was supported by NSF through award #1433763

*Acknowledgements.*    We thank Walter Robinson, Carl Schreck, Arlene Laing, James Russell, and Ademe Mekonnen for useful discussions.
The original code for the model used here was kindly provided by Nick Hall.





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
