# Peer review of "African Easterly Waves in an Idealized General Circulation Model: Instability and Wavepacket Diagnostics"

_Weather and Climate Dynamics, 2020_

## Referee Comment (RC1) · Anonymous Referee #1 · 12 Nov 2020

Recent studies have suggested that African easterly waves are not sufficiently accelerated by baroclinic and barotropic instabilities to explain the amplitudes the waves achieve as they propagate along the African easterly jet. The previous works have shown that initial disturbance by forcing from extratropical waves or by bursts of convection initiate the waves. These authors argue, in contrast, that baroclinic and barotropic instabilities act along clusters of waves instead of just to amplify an individual wave. They suggest that group velocities of the waves, corrected for quasi Doppler shifting, can maintain wave packets in the jet for extended periods of time, thereby allowing these instabilities to contribute substantially to the wave amplitude. The authors use reanalysis data, presented in Figure 1, to motivate the argument that eddy dispersion

can move the clusters of waves eastward or westward within the jet domain, making room for their arguments. They argue that a wave packet that has net zero group velocity somewhere in the packet would allow the wave pack to exist in the same region for extended periods of time. They suggest that a slowly varying background flow would allow for slow adjustment between westward, eastward, and stationary group propagation that would yield periods of enhanced or reduced easterly wave activity.

I think their arguments are generally well-made, and their model analysis compelling. Their results generally support their conclusions. They also include important caveats, such as the relevance of moist processes or interactions with other modes, which their model experiments do not accommodate. My primary concern is that the authors apply similar caveats to their interpretation of Figure 1. Figure 1 is included as simple observational support for the idea that clusters of waves can move eastward, westward, or be stationary in the African easterly jet region. They infer from this figure that wave group velocities behave similarly. Although I agree from first principles that the authors' point is likely true (i.e., given that the jet is roughly 5 m/s westward, and that somewhat similar mixed Rossby gravity waves have an eastward group velocity around 5 m/s eastward, it is conceivable, as a back of the envelope estimate, that easterly wave packets could become trapped in the jet region). I refer here to MRG waves just as an illustration, since we have a better understanding of their fundamental group velocity characteristics. Since the time Wheeler and Kiladis (1999) showed that observed MRG waves do not conform to theoretical MRG wave group velocities, authors have continued to attempt to estimated group velocities from longitude time diagrams of observed data. The problem with attempting such estimation is that amplitude along an observed wave packet is impacted by many other mechanisms beyond group propagation. Consider the illustration of an MRG wave packet interacting with the MJO. The group velocity of mixed Rossby gravity (MRG) waves can be close to the phase speed of the MJO, but let's assume here that MRG wave group velocity is 1 m/s slower than the MJO phase speed. The MJO is well known to modulate the amplitude of MRG waves. Now, assume that the MJO is moving from east to west across the

packet, transitioning from the active convective phase to the suppressed convective phase. Since the MJO, in this case, is faster than the MRG packet, but since the MJO will be preferentially amplifying the eastern side of the packet relative to the western side, interaction with the MJO will cause the wave to appear to disperse eastward more rapidly than would have been predicted in the absence of the MJO. Yet this signal is not true dispersion, but is a nonlinear process caused by the MJO modulation of the MRG convection, which, in turn, feeds back on MRG circulation. Interaction with other waves and changing base state signals would window activity in easterly waves in a manner that would mimic dispersion, even if easterly waves were not fundamentally dispersive (herein I'm not arguing that point is true). Further, we know that breaking extratropical waves can force new easterly waves. The longitude at which the extratropical waves break can adjust with time, eastward or westward, thereby making the resultant waves appear to have eastward or westward group velocities. I suggest, therefore, that the authors revise their discussion of Figure 1 to include these caveats. Their point is well taken that observations show the waves clustering sometimes toward the west and sometimes toward the east. It is just not clear from these figures that this outcome is consistent with the waves' actual group velocities. The authors' model experiments appear robust to these mechanisms, but they need to caveat that their results do not show how important the effects they demonstrate are in comparison to effects like the ones I describe above.

---

## Referee Comment (RC2) · Anonymous Referee #2 · 4 Dec 2020

The study investigates the evolution of wave packets that develop in different basic states in an idealised GCM. Those basic states include the climatology in JJAS between 1987 and 2017, 755 individual basic states based on 15-day averages of the JJAS data, and three averages based on the lifetime of the wavepackackets in the 755 basic states. The study found that a localized AEJ supports downstream and upstream dispersive wavepackets. The authors point out that the response is independent of how the waves have been triggered. They also show that the wavepackets remain withing the AEJ and are not swept away. This study builds up on previous work by Hall et al. (2006), Thorncroft et al. (2008) and Leroux and Hall (2009), using a very similar model set up to allow for direct comparison to the previous studies. The authors include the

effects of moist convection and dust radiative forcing in their experiment to study the wavepacket response. The results of this study provide inside into the dynamics of the AEJ and AEWs and will be of interest to the community.

Minor comments:

1. P. 2, l.45 and p.3, l. 62: It is not clear what you mean by "antifriction" in this context. Could you use a different word or provide a bit more explanation in the text?

2. P. 3, l. 60: What do you mean my "the background PV gradient is single-signed"? Do you mean by that that there is no meridional reversal of the PV gradient?

3. P. 3, l. 78: "s" does not need to be in italics.

4. Figure 1 and others: Why don't you call the time-longitude diagrams Hovmoeller plots? You do refer to them later in the text. They are a common type of diagram so why not be consistent throughout? Also, in Latex you can use the correct way of writing with two dots on the "o", which I can't do in Word.

5. Figure 1: The caption would be easier to read if you move (a) before "the year 2006" and (b) before "2008". You use this order of giving the labels of the subplots in some of the later figure captions. It would be good to be consistent. I think ERAI is more commonly abbreviated with a capital "I".

6. P. 2, l. 94: "the wavepackets are collocated with the AEJ" – Are you referring to the right column of Figure 6? However, you are still talking about Fig. 1 and I can't see the location of the AEJ. Could you potentially add the jet location using contours?

7. P. 4, l. 93: Avoid using "significant" if you haven't carried out any significance tests.

8. P. 4, l. 97: Remove space before the full stop.

9. P. 4: l. 99: Include "relation" or something similar after "this".

10. General comment: You start a number of sentences with "this" also in consecutive

sentences. It is not always clear what "this" refers to and makes the sentences a bit unspecific. That could be avoided by adding, for instance, a noun after "this".

11. P. 4, l. 101: What does "its" refer to? The wavepacket?

12. P. 6, l. 137: Remove "also".

13. P. 7, l. 155: What do you mean by "this posits"?

14. P. 7, l. 179: Hemispheres – plural needed here.

15. P. 8, l. 202, 204: Comma needed after the equation.

16. P. 8, l. 208-208: What do you mean by "it"?

17. P. 8-9: l.211-220: After the dash start with an upper-case letter. I think it is a bit confusing that you give the total number of basic states of 779 before you explain how you create them. Could you say "in the following three types of basic states:" in line 201 and then insert after the itemization something like "In total that gives 779 basic states".

18. P. 9, l. 222: "Each simulation". I'm not really sure how you run these simulations. Do you drive the GCM with each of the basic states? Could you make that clearer in the text?

19. Fig. 3: From the text it is clear that the figure is based on the climatological basic state, but that is not clear from the figure caption.

20. Figs. 4, 7: Add "horizontal" before "wind".

21. P. 10, l 229: "fixed heating produces a baroclinic vortex" – Where and why? A bit more explanation would be good here.

22. P. 10. L. 235, p. 13, l. 266, p. 14, l. 273: Delete "clearly" and in other places too. "Clear" and "clearly" can mean different things to different people.

23. Fig. 5: Why does panel (a) have no colour bar? Or is it the same as in (b)? That

is not obvious. The black dot here and in other figures is a bit hard to see and it looks more like a half circle than a dot.

24. Fig. 6 and a few other of the following figures: The labels are too small to read. Is that because of the resolution of the figure? Is that lower for the review process than for publication? It would be better to make sure that labels can be read easily.

25. P. 12, l. 246: When talking about surface westerlies, please refer to the right column of Fig. 6. I assume this is where you want the reader to look at.

26. P. 12, l. 250: "Consistent with it" – "it" refers to what?

27. Section 4.2: Why is the behaviour so different for the long-lived basic stated compared to the short-lived and intermediate basic state?

28. P. 14, l. 283: Is there no commonly used symbol for the growth rate? A dot is missing at the end of the sentence.

29. P. 14, l. 292: p and A need to be defined as well.

30. P. 16, l 297: Perturbation velocity has already been defined. Remove the space before the comma.

31. P. 14, l. 300: What do you mean by "half-wavelengths in the zonal direction"?

32. P. 14, l. 303-304: Remove the space before the degree symbols.

33. Fig. 13. Here the labels are too small again. In the bottom row the labels seem to be partially cut off. Labelling the subfigures with a, b, c, d would be good. In the caption you refer only to panel (a) and not the others.

34. P. 17, l. 307: Lower-case "in".

35. P. 17, l. 309: What do you mean by "these additional energy sources"? Are you referring to the destabilizing role of moist convection and SMD?

36. P. 17, l. 316 – 317. Three sentences begin with "this". Particularity the last "this"'

is unspecific.

37. Fig. 14: Levels are cut off on the y-axis of panel (a). Lower-case 's' for steamfunction and remove the space before the degree symbol.

38. Fig. 15: The labels are too small.

39. P. 18, l. 321: Sometimes you reference equations as Eq. X and sometimes as Equation X. Please be consistent.

40. Fig 16: Panels need labels.

41. P. 19, l. 328: Comma after "however".

42. P. 19, l. 330: Comma after "case" and delete "this is".

43. P. 19, l. 335: Not clear what you mean by "this" here. Do you mean your study or analysis?

44. P. 19, l. 343: What do you mean "three additional simulations are performed"?

45. P. 20, l. 347: PV has already been defined, so please use it.

46. P. 20, l. 358: Insert "located" before "above".

47. P. 20, l. 258-355: When you say stronger and higher please add compared to what. Consider adding a comma after "conversions" and replacing "this" with "which".

48. P. 21, l. 379: You could remind the reader what you mean by "criticism regarding the limited zonal extent of the AEJ" as this is one of your main points.

49. P. 21, l. 383: Do you mean your results? What does "this" refer to? There is another "this" in the next sentence.

50. P. 21, l. 395-396: The hyphens have a different length.

51. The list of references contains inconsistencies. For some papers the paper title is written so that every first letter of the word is a capital letter, but for some papers that

is not the case.

---

## Referee Comment (RC3) · Anonymous Referee #3 · 16 Dec 2020

Review of

**African Easterly Waves in an Idealized General Circulation Model:**
**Instability and Wave Packet Diagnostics**
by
**Joshua White and Anantha Aiyyer**

**RECOMMENDATION: Accept** after only minor revisions.

**OVERVIEW**

This is an interesting study that advances understanding of the evolution of African easterly wave (AEW) packets in zonally varying African easterly jets (AEJs). At the heart of the study is the examination of how the zonal extent of the AEJ affects the growth of the AEWs. This study builds on prior work by Hall et al. (2006), Thorncroft (2008), and Leroux and Hall (2005), who showed that in a zonally varying AEJ, realistic boundary layer damping suppresses, and even reverses, the normal mode growth rates of the AEWs. These studies, however, did not consider how the zonal extent of the AEJ affected the growth rates and longevity of the AEWs.

To carry out the analysis, the authors employ an idealized general circulation model with realistic boundary layer damping; the basic states are constructed from global reanalysis fields. The central result of the numerical results is that irrespective of the length of the AEJ, the AEW packets grow and remain confined to the AEJ, which is reminiscent of absolute instability, as discussed heuristically by the authors. The authors find that for most basic states the AEWs are damped, though for some basic states the AEWs are able to grow exponentially. The connection between the lifetimes of the packets, which are distinguished as short, medium and long based on criteria defined by the authors, provides further insights into the AEJ-AEW connection.

The hypotheses that motivate the study are clear, the experiments are well designed, and the analyses are comprehensive.

**COMMENTS**

**Comment A:** It seems to me that it would be instructive to have a quantitative measure for the limited zonal extent of the AEJ. The authors could then quantitatively compare its zonal extent with the amplification and longevity of the AEWs. In fact, dynamically, comparison between the potential vorticity (PV) gradient and the longevity of the waves would be the most insightful. Say, for example, the meridional gradient of PV changes sign in a zonally limited section of the jet. How does the zonal length of the sign reversal region relate to the longevity of the waves? Perhaps the authors could comment on this.

**Lines 45:** The term "antifriction" is a bit unclear, although this term is commonly used in engineering when referring to lubricants in bearings, for example. I am unaware of it being used in an atmospheric context. Perhaps the subsection heading could be: *The destabilizing influence of moist convection and dust aerosol loading*.

**Line 47:** The first sentence of the paragraph states: "…two critical aspects of the dynamics are missing in their simulations." It might be clearer to start the next sentence as: "First, their model has no…" Then the next paragraph, **line 55**, could perhaps start with: The other aspect that was not considered in the simulations of Hall et al. (2006) and Hall (2009) were the feedbacks associated with aerosol loading…

**Line 153:** Perhaps make this question a little clearer by writing: What mediates the vacillation of AEW activity?

**Lines 154-155:** When the authors state the slowly varying background flow alternates aperiodically…" do they mean that it alternates aperiodically in time or space, or both. Also, how do the authors define "slowly varying?

**Comment B:** For my eyes, the figures are too small. Perhaps they can be enlarged so that the axis notation and wind contours are more easily seen.

**Line 238:** The authors state: "Importantly, as seen in observations, the AEW wavepacket…" It would be helpful if a couple of references could be provided regarding the observations. Also, "AEW wavepacket" should perhaps be "AEW packet", here and elsewhere.

**Line 347:** "…basic states considered here are associated with reversal in potential vorticity gradients,…" Are the reversal in PV gradients over the entire zonal extent of the jet or only over a portion? In either case, what are the physical implications?

---

## Author Comment (AC1) · 31 Jan 2021

We thank the reviewer for their time and effort.

The reviewer makes an important point regarding our interpretation of Fig. 1. Since it is derived from global reanalysis – our proxy for observations – it is important to acknowledge that the signature seen in the figure is not guaranteed to be a pure AEW packet that is evolving independently of external influences. Indeed, as the reviewer points out, the modulation of AEWs by the Madden-Julian Oscillation, convectively coupled Kelvin waves, and breaking extratropical waves can give the appearance

of a dispersing wavepacket if there is a preferential amplification of one part of the packet compared to the other. As recommended, we have now added a paragraph that explicitly states the caveat in our interpretation in section 2.1.

Furthermore, our numerical simulations produce AEW packets that decay or grow monotonically. Per the construct of our experiments, there is no possibility of the modulation of their growth by external sources listed above. It is important to document the impact of interactions with externally imposed sources of wave forcing on the group dynamics of AEWs. This is an area of further investigation that will be reported separately. We have now added text that explicitly states the caveat in our interpretation in section 5.

**1 Specific Edits made**

1. The following text has been included in Section 2.1 of the revised paper:

   An important caveat should be recognized in relation to Fig. 1. We have interpreted it as a pure AEW packet. In nature, however, a variety of tropical and extratropical systems ranging from synoptic (e.g., equatorial waves, breaking extratropical waves) to intraseasonal (the Madden-Julian Oscillation) can modulate the amplitude of AEWs (e.g. Matthews, 2004; Leroux and Hall, 2009; Ventrice et al., 2011; Alaka and Maloney, 2012, 2014). This modulation could, in principle, present itself like the dispersion of a linear wavepacket if it leads to preferential amplification of one side of the packet. In a related issue, Aiyyer et al. (2012) showed that cloud signatures associated with tropical cyclones can artificially project onto a wide range of eastward and westward propagating equatorial

modes as a result of the filtering in the wavenumber-frequency domain. The use of idealized numerical models, where the primary response in the model is the AEW stormtrack, mitigates some of this concern and provides an independent assessment of the relevance of group dynamics for observed AEW packets.

2. The following text has been included in Section 5 (Discussion) of the revised paper:

In addition, our simulations do not include the modulation of AEWs by external phenomena such as the Madden Julian Oscillation, convectively coupled Kelvin waves, or breaking extratropical baroclinic waves that are commonplace in nature. The impact of these externally imposed source of wave forcing on the group dynamics of AEWs also needs to be examined in future studies.

---

## Author Comment (AC2) · 1 Feb 2021

We thank reviewer 3 for their time and effort.

**1   Responses to comments**

Please see below our responses to specific comments. The original reviewer comments are included.

[Figure]

Comment A: It seems to me that it would be instructive to have a quantitative measure for the limited zonal extent of the AEJ. The authors could then quantitatively compare its zonal extent with the amplification and longevity of the AEWs. In fact, dynamically, a comparison between the potential vorticity (PV) gradient and the longevity of the waves would be the most insightful. Say, for example, the meridional gradient of PV changes sign in a zonally limited section of the jet. How does the zonal length of the sign reversal region relate to the longevity of the waves? Perhaps the authors could comment on this.

**Response** We agree, a quantitative measure for the limited zonal the extent of the AEJ will be useful. We have now referred to both Molinari and Dickinson (2000) and Thorncroft et al.(2008) to make the point about the zonal extent of the AEJ. The former, in particular, clearly shows that the meridional gradient reversal of PV associated with the AEJ during July–October spans around 60–70$^o$ longitudes. Assuming AEW wavelengths around 2000-4000 km, this corresponds to aboCharney and Stern's (1962) and Fjortoft's (1950) ut 2-3 wavelengths at most, consistent with Thorncroft et al.(2008).

We have not examined the relationship between the growth rate and the characteristics of the background environment beyond classifying all simulations into short, intermediate and long-lived categories. As the reviewer points out, one possible option is to compare the growth rates and metrics of the African easterly jet (AEJ) such as the zonal extent and the strength of the potential vorticity (PV) gradients. There are two main reasons.

First, the concern was to focus on the group dynamics that lead to these starkly different outcomes. The point that we make is that despite these different outcomes, nearly all AEW packets appear to be nearly stationary. The point that we make is that, since the packet is not swept out and damped rapidly, there is more likelihood of coupling with convection and dust radiative effects that could account for the existence of AEWs in nature.

Second, Leroux and Hall (2009) attempted to relate wave growth to the strength of the AEJ and PV reversal. They did not find a clear signature of the impact of these parameters. On the other hand, they found that the surface area covered by these two parameters was a better indicator of wave growth. In light of this, as well as recognizing that any rigorous attempt at accounting for wave growth should include moist convection and dust radiative effects, we did not pursue this avenue of inquiry. We do plan to examine this issue using a model that can represent these additional diabatic effects.

Lines 45: The term "antifriction" is a bit unclear, although this term is commonly used in engineering when referring to lubricants in bearings, for example. I am unaware of it being used in an atmospheric context. Perhaps the subsection heading could be: The destabilizing influence of moist convection and dust aerosol loading.

**Response** Thanks for the suggestion. We have reworded the heading to: "Destabilization by moist convection and dust aerosol forcing."

Line 47: The first sentence of the paragraph states: "...two critical aspects of the dynamics are missing in their simulations." It might be clearer to start the next sentence as: "First, their model has no..." Then the next paragraph, line 55, could perhaps start with: The other aspect that was not considered in the simulations of Hall et al. (2006) and Hall (2009) were the feedbacks associated with aerosol loading...

**Response** Thanks for the suggestion. We have done that.

Line 153: Perhaps make this question a little clearer by writing: What mediates the vacillation of AEW activity?

**Response** Thanks for the suggestion. We have done that.

Lines 154-155: When the authors state the slowly varying background flow alternates aperiodically..." do they mean that it alternates aperiodically in time or space, or both. Also, how do the authors define "slowly varying?

**Response** We have clarified that it alternates aperiodically in time. The flow is also fully varying in space. The slow variation refers to the running 15-day average fields used for basic states. This ensures a slow, but steady change in the basic states. By considering 775 basic states constructed thus, we get a clearer picture of the AEW packet behavior as compared to one climatological basic state.

Comment B: For my eyes, the figures are too small. Perhaps they can be enlarged so that the axis notation and wind contours are more easily seen.

**Response** All figures have been enlarged.

Line 238: The authors state: "Importantly, as seen in observations, the AEW wavepacket..." It would be helpful if a couple of references could be provided regarding the observations. Also, "AEW wavepacket" should perhaps be "AEW packet", here and elsewhere.
**Response** We have now provided a reference to Fig. 1 in this paper and also external citations to Diaz and Aiyyer (2013a, 2015). We have also replaced all instances of AEW wavepacket with AEW packet.

Line 347: "...basic states considered here are associated with a reversal in potential vorticity gradients,..." Are the reversal in PV gradients over the entire zonal extent of the jet or only over a portion? In either case, what are the physical implications?

**Response** The reversal in basic state PV typically coincides with the AEJ. When averaged over several days, this appears as nearly continuous over the length of the AEJ. The climatological structure of the PV fields can be seen in Dickinson and Molinari (2000) and Russell and Aiyyer (2020). The reversal in PV gradient, along with the sign of the zonal flow over North/West Africa satisfies necessary conditions for hydrodynamic instability (Charney and Stern 1962; and Fjortoft 1950). This was first shown by Burpee (1972).

---

## Author Comment (AC4) · 2 Feb 2021

We are grateful to reviewer 2 for their time and effort in helping improve the content of the paper.

**1 Responses to comments**

1. P. 2, l.45 and p.3, l. 62: It is not clear what you mean by "antifriction" in this context.

[Figure]

Could you use a different word or provide a bit more explanation in the text?

**Response:** We have replaced this by: Destabilization by moist convection and dust aerosol forcing both in the section title as well as in the paragraph.

2. P. 3, l. 60: What do you mean my "the background PV gradient is single-signed"? Do you mean by that that there is no meridional reversal of the PV gradient?

**Response:** Yes.

3. P. 3, l. 78: "s" does not need to be in italics.

**Response:** Thanks! Good catch.

4. Figure 1 and others: Why don't you call the time-longitude diagrams Hovmoeller plots? You do refer to them later in the text. They are a common type of diagram so why not be consistent throughout? Also, in Latex you can use the correct way of writing with two dots on the "o", which I can't do in Word.

**Response:** Done!

5. Figure 1: The caption would be easier to read if you move (a) before "the year 2006" and (b) before "2008". You use this order of giving the labels of the subplots in some of the later figure captions. It would be good to be consistent. I think ERAI is more commonly abbreviated with a capital "I".

**Response:** Corrected.

6. P. 2, l. 94: "the wavepackets are collocated with the AEJ" – Are you referring to the right column of Figure 6? However, you are still talking about Fig. 1 and I can't see the location of the AEJ. Could you potentially add the jet location using contours?

**Response:** We have added the reference to Figure 3 for the location of the jet to make this clear.

7. P. 4, l. 93: Avoid using "significant" if you haven't carried out any significance tests.
**Response:** Agreed. Replaced it by "much."

8. P. 4, l. 97: Remove space before the full stop.

**Response:** Corrected.

9. P. 4: l. 99: Include "relation" or something similar after "this".

**Response:** Done

10. General comment: You start a number of sentences with "this" also in consecutive sentences. It is not always clear what "this" refers to and makes the sentences a bit unspecific. That could be avoided by adding, for instance, a noun after "this".

**Response:** We have attempted to do that where ever the context was not clear

11. P. 4, l. 101: What does "its" refer to? The wavepacket?

**Response:** Yes

12. P. 6, l. 137: Remove "also".

**Response:** Done

13. P. 7, l. 155: What do you mean by "this posits"?

**Response:** We mean that this asserts or postulates

14. P. 7, l. 179: Hemispheres – plural needed here.

**Response:** Done

15. P. 8, l. 202, 204: Comma needed after the equation.

**Response:** Done

16. P. 8, l. 208-208: What do you mean by "it"?

**Response:** it refers to $\beta$

17. P. 8-9: l.211-220: After the dash start with an upper-case letter. I think it is a bit confusing that you give the total number of basic states of 779 before you explain how you create them. Could you say "in the following three types of basic states:" in line 201 and then insert after the itemization something like "In total that gives 779 basic states".

**Response:** Thanks! Done.
18. P. 9, l. 222: "Each simulation". I'm not really sure how you run these simulations.Do you drive the GCM with each of the basic states? Could you make that clearer in the text?

**Response:** Yes, we do. The GCM is run 779 different times. Each run has a different basic state. This is now explicitly mentioned in Section 3.2.

19. Fig. 3: From the text it is clear that the figure is based on the climatological basic state, but that is not clear from the figure caption.

**Response:** We have added "JJAS 1987–2017 averaged" to the beginning of the figure caption.

20. Figs. 4, 7: Add "horizontal" before "wind".

**Response:** Corrected

21. P. 10, l 229: "fixed heating produces a baroclinic vortex" – Where and why? A bit more explanation would be good here.

**Response:** We have now clarified that this is consistent with the results shown in Thorncroft et al. (2008). Since the analysis of the transient response is not the focus here, we hope that the interested reader will refer to Thorncroft et al. (2008).

22. P. 10. L. 235, p. 13, l. 266, p. 14, l. 273: Delete "clearly" and in other places too. "Clear" and "clearly" can mean different things to different people.

**Response:** Agreed. Done

23. Fig. 5: Why does panel (a) have no colour bar? Or is it the same as in (b)? That is not obvious. The black dot here and in other figures is a bit hard to see and it looks more like a half circle than a dot.
**Response:** The label bar was originally left off since scale is not important with our linear model, but we have added the label bar to Fig 5(a) to be consistent. Also, we have changed "black dot" to "black semicircle at time 0 days"

24. Fig. 6 and a few other of the following figures: The labels are too small to read. Is that because of the resolution of the figure? Is that lower for the review process than for publication? It would be better to make sure that labels can be read easily.

**Response:** The figures have been enlarged.

25. P. 12, l. 246: When talking about surface westerlies, please refer to the right column of Fig. 6. I assume this is where you want the reader to look at.

**Response:** Done

26. P. 12, l. 250: "Consistent with it" – "it" refers to what?

**Response:** In the context of the preceding sentence, "it" refers to the reversal in PV gradient which along with the sign of the zonal flow, satisfies the condition for mixed barotropic-baroclinic instability.

27. Section 4.2: Why is the behaviour so different for the long-lived basic stated compared to the short-lived and intermediate basic state?

**Response:** As we point out in the discussion, in all basic states, a near-stationary wavepacket ensues. Each state is unstable under inviscid conditions. Inclusion of damping, however effectively stabilizes the short and intermediate basic state. For the long-lived case, the energy conversions are sufficiently strong to over come reasonable damping.
28. P. 14, l. 283: Is there no commonly used symbol for the growth rate? A dot is missing at the end of the sentence.

**Response:** Added the dot.

29. P. 14, l. 292: p and A need to be defined as well.

**Response:** Done.

30. P. 16, l 297: Perturbation velocity has already been defined. Remove the space before the comma.

**Response:** Done.

31. P. 14, l. 300: What do you mean by "half-wavelengths in the zonal direction"? **Response:**

To calculate the surface integral, we define the area A such that it spans half the wavelength in the east-west direction and 5–30N in the north-south direction. This is based on the work of Orlanski and Chang (1993) as adapted by Diaz and Aiyyer (2015).

32. P. 14, l. 303-304: Remove the space before the degree symbols.

**Response:** Done.

33. Fig. 13. Here the labels are too small again. In the bottom row the labels seem to be partially cut off. Labelling the subfigures with a, b, c, d would be good. In the caption you refer only to panel (a) and not the others.

**Response:** We have increased the label size and fixed the bottom row labels that were partially cut off, and we have labeled the subfigures and called them out appropriately in the caption.

34. P. 17, l. 307: Lower-case "in".

**Response:** Done.

35. P. 17, l. 309: What do you mean by "these additional energy sources"? Are you referring to the destabilizing role of moist convection and SMD?

**Response:** Yes.

36. P. 17, l. 316 – 317. Three sentences begin with "this". Particularity the last "this"' is unspecific.

**Response:** Agreed, thanks. We have reworded and edited this section to make it clearer.

37. Fig. 14: Levels are cut off on the y-axis of panel (a). Lower-case 's' for steamfunction and remove the space before the degree symbol.

**Response:** Done.

38. Fig. 15: The labels are too small.

**Response:** Label sizes have been increased.

39. P. 18, l. 321: Sometimes you reference equations as Eq. X and sometimes as Equation X. Please be consistent.

**Response:** Done.

40. Fig 16: Panels need labels.

**Response:** Labels have been added.

41. P. 19, l. 328: Comma after "however". **Response:** Done.

42. P. 19, l. 330: Comma after "case" and delete "this is".

**Response:** Done.

43. P. 19, l. 335: Not clear what you mean by "this" here. Do you mean your study or analysis?

**Response:** Yes, the paper.

44. P. 19, l. 343: What do you mean "three additional simulations are performed"?

**Response:** It refers to the 3 ensemble averaged basic states. This sentence has been reworded.

45. P. 20, l. 347: PV has already been defined, so please use it.

**Response:** Done.

46. P. 20, l. 358: Insert "located" before "above".

**Response:** Done.

47. P. 20, l. 258-355: When you say stronger and higher please add compared to what.Consider adding a comma after "conversions" and replacing "this" with "which".

**Response:** The comparison is relative to the short-lived case as noted in the previous sentence. We have edited the sentence per your recommendation.

48. P. 21, l. 379: You could remind the reader what you mean by "criticism regarding the limited zonal extent of the AEJ" as this is one of your main points.

**Response:** Done. The sentence now reads: This addresses the criticism that the limited zonal extent of the AEJ may be an impediment to AEW growth.

49. P. 21, l. 383: Do you mean your results? What does "this" refer to? There is another "this" in the next sentence. **Response:** Thanks again for pointing this out. We have edited

the sentences for clarity.

50. P. 21, l. 395-396: The hyphens have a different length.

**Response:** Fixed it.

51. The list of references contains inconsistencies. For some papers the paper title is written so that every first letter of the word is a capital letter, but for some papers that is not the case.

**Response:** We used the WCD LATEX template for typesetting and will work with the copy editors to fix this.

---

## Author Response (AR2)

**Responses to co-editor's comments**

Joshua White[1] and Anantha Aiyyer[1]

[1]Department of Marine, Earth and Atmospheric Sciences, North Carolina State University

**Correspondence:** A. Aiyyer (aaiyyer@ncsu.edu)

We are grateful to the co-editor for examining our responses to the reviewer comments and for suggesting edits. We have corrected the typographical errors suggested by the co-editor.